# Tryptophan-Derived Metabolites and Glutamate Dynamics in Fatal Insulin Poisoning: Mendelian Randomization of Human Cohorts and Experimental Validation in Rat Models

**DOI:** 10.3390/ijms26094152

**Published:** 2025-04-27

**Authors:** Yuhao Yuan, Yu Liu, Shengnan Wang, Jiaxin Zhang, Xiangting Gao, Yiling Li, Zhonghao Yu, Yiwu Zhou

**Affiliations:** Department of Forensic Medicine, Tongji Medical College, Huazhong University of Science and Technology, Wuhan 430000, China; d202381918@hust.edu.cn (Y.Y.); fyliuyu@outlook.com (Y.L.); wangshengnan@hust.edu.cn (S.W.); d202181694@hust.edu.cn (J.Z.); d202482115@hust.edu.cn (X.G.); m202275691@hust.edu.cn (Y.L.)

**Keywords:** hypoglycemic encephalopathy, mendelian randomization, metabolomics, tryptophan metabolism pathway, forensic identification

## Abstract

Insulin overdose may cause hypoglycemic encephalopathy. In this study, Mendelian randomization was employed to analyze changes in the serum metabolites of patients with hypoglycemic encephalopathy, and metabolomics analysis was conducted to detect differential metabolites in the serum of a rat model of hypoglycemic encephalopathy induced by insulin overdose. The results indicated an overall upward trend in the tryptophan metabolism pathway in patients with hypoglycemic encephalopathy and rats with hypoglycemic encephalopathy caused by insulin overdose, while serum glutamate levels declined. The metabolic changes in the tryptophan pathway provide new insights into the impact of hypoglycemia on brain function. The related products of the tryptophan metabolism pathway have a certain diagnostic value for hypoglycemic encephalopathy and forensic identification of insulin overdose-induced hypoglycemic encephalopathy death.

## 1. Background

Human insulin is synthesized by pancreatic β-cells and plays a critical role in regulating the metabolism of carbohydrates, lipids, and proteins in the body [1]. Currently, insulin remains the most effective therapeutic agent for diabetes mellitus, with widespread use and accessibility [2]. However, hypoglycemia induced by insulin overdose remains a clinical concern [2,3]. In the United States alone, approximately 100,000 individuals are hospitalized annually due to insulin-induced hypoglycemia [4], and severe hypoglycemic episodes can be fatal [5,6]. Notably, a growing number of homicide cases using insulin as a tool for homicide have been reported [6,7,8]. Exogenous insulin injection has even been described as the “perfect murder weapon” [9,10]. Hypoglycemia caused by insulin overdose is a primary factor contributing to multi-organ damage and mortality [9], with hypoglycemic encephalopathy being the most severe manifestation [11,12].

Hypoglycemic brain injury refers to reversible or irreversible damage in specific brain regions resulting from prolonged cerebral energy deprivation under low blood glucose conditions (flat electroencephalogram [EEG] waves lasting ≥ 30 min) [13]. The brain requires substantial energy in the form of a constant glucose supply as the central nervous system (CNS) cannot synthesize glucose autonomously. Hypoglycemia disrupts CNS energy homeostasis, impairing normal physiological function. Severe hypoglycemia induces irreversible damage in vulnerable brain regions, such as cortical layers 2–3, the hippocampal CA1 region, and the dentate gyrus [14]. Concurrently, systemic metabolic alterations—including amino acid and lipid metabolism—occur during hypoglycemic encephalopathy [15].

Based on these findings, this study was conducted to investigate insulin-induced hypoglycemic encephalopathy. Mendelian randomization was employed to analyze changes in serum metabolites in patients with hypoglycemic encephalopathy. Additionally, metabolomics methods were utilized to detect differential metabolites in the serum of a rat model of hypoglycemic encephalopathy induced by insulin overdose. The present study aimed to identify common changes in metabolites and metabolic pathways between the two groups. Investigating metabolic alterations in hypoglycemic encephalopathy provides evidence supporting clinical diagnosis and treatment while also offering valuable forensic insights into cases related to insulin overdose fatalities.

## 2. Results

### 2.1. Causal Links Between Hypoglycemic Encephalopathy and Plasma Metabolites

This Mendelian randomization (MR) study investigated potential causal relationships between 871 plasma metabolites and hypoglycemic encephalopathy (HE). Genome-wide significant SNPs were selected as instrumental variables (IVs), with 13 SNPs meeting stringent criteria (Appendix A). The high F-statistic (>10) of these IVs confirmed their robustness, thereby minimizing bias and strengthening causal inference.

An inverse variance weighted (IVW) analysis identified 34 metabolites significantly associated with HE. Five metabolites, including isovalerylglycine, trans 3,4-methyleneheptanoate, free proline, glutamate, and palmitoylcarnitine, showed inverse correlations with HE. In contrast, 29 metabolites exhibited positive associations, including pregnenediol disulfate, palmitoyl sphingomyelin, taurine, kynurenine, and tryptophan (full list provided in Figure 1).

### 2.2. Sensitivity Validation

Sensitivity analyses were conducted using Cochran’s Q-test, MR-Egger regression, and MR-PRESSO to evaluate horizontal pleiotropy. Among the 34 metabolites, only isovalerylglycine displayed heterogeneity via IVW (*p* = 0.035), while MR-Egger detected no heterogeneity. After excluding isovalerylglycine, the 33 remaining metabolites showed consistent associations (Figure 2). MR-Egger intercept tests and MR-PRESSO global analyses revealed no horizontal pleiotropy. Leave-one-out sensitivity analysis further validated the stability of these associations, as the iterative exclusion of individual SNPs produced effect estimates consistent with the primary MR results. Pathway enrichment analysis was deferred to the experimental cohort due to the limited number of MR-significant metabolites in the human cohort (n = 33) and the prioritization of cross-species validation.

### 2.3. General Condition of Hypoglycemic Encephalopathy Rats

Within 1.5–2.0 h following intraperitoneal insulin administration, the blood glucose levels of the rats gradually decreased to below 2.5 mmol/L. The initial symptoms included limb weakness, reduced activity, vocalization, and labored breathing. As hypoglycemia progressed, the rats exhibited severe manifestations such as incontinence, convulsions, loss of righting reflex, generalized rigidity, and opisthotonos. Following ventilator-assisted anesthesia, electroencephalogram (EEG) recordings in insulin-overdosed rats transitioned from α-waves to slower θ- and δ-waves with higher amplitudes. Thereafter, epileptiform discharges (e.g., polyspikes, rhythmic sharp waves, spike-and-slow wave complexes) were observed, accompanied by myoclonic jerks and seizures. Moreover, burst suppression patterns were found in some rats, culminating in isoelectric periods (Figure 2A).

The pre- and post-intervention blood glucose dynamics are shown in Figure 2B. Baseline glucose levels showed no significant differences between groups. Following the intervention, insulin administration caused a marked decrease in glucose levels in the experimental group, while sham procedures induced stress-related hyperglycemia in controls. The control rats underwent sham operations under isoflurane anesthesia to account for surgical stress, which may have transiently elevated blood glucose via catecholamine release.

### 2.4. Data Quality Control

All samples, including quality control (QC) replicates, were analyzed for small-molecule metabolites via UPLC-MS/MS. Base peak chromatograms (BPCs) were generated by plotting the highest ion intensity at each time point, with retention time on the *x*-axis and signal intensity on the *y*-axis. Furthermore, the overlaid BPCs of QC samples in the ESI+ and ESI− modes (Figure 3 and Appendix A) demonstrated robust chromatographic separation and retention time alignment, confirming the system’s stability.

Principal component analysis (PCA) of all samplesrevealed tight clustering of QC replicates without outliers, indicating minimal technical variability and high reproducibility.

### 2.5. Multivariate Statistical Analysis

PCA modeling of the experimental and control groups (Figure 4A) showed clear separation along the first principal component (PC1), with red and green points representing control and experimental biological replicates, respectively. Both groups clustered within 95% confidence ellipses, displaying no outliers.

Orthogonal partial least squares-discriminant analysis (OPLS-DA) is a supervised method, showing maximized inter-group discrimination on the horizontal axis while vertically isolating intra-group variability (Appendix A). The model exhibited strong predictive validity (R2Y = 0.956, Q2 = 0.903), with variable importance in the projection (VIP) scores guiding biomarker selection.

### 2.6. Univariate Statistical Analysis

A total of 1875 small-molecule metabolites were identified by UPLC-MS/MS, with 374 endogenous metabolites annotated via the KEGG and HMDB databases. Differential metabolites (n = 502) were filtered using thresholds of VIP ≥ 1, q-value < 0.05, and fold change (FC) ≥1.5 or ≤0.67. Among these, 167 were upregulated and 335 were downregulated, including 67 endogenous metabolites (23 upregulated, 44 downregulated) such as amino acids, fatty acids, carbohydrates, and bile acids.

Volcano plots (Figure 4B) were used to visualize log2 (FC) versus −log10 (q-value), with red/green points denoting significantly upregulated/downregulated metabolites. Endogenous differential metabolites are detailed in Appendix D. The metabolite identification details are provided in Appendix A.

Hierarchical clustering heatmaps of endogenous differential metabolites (Figure 4C) display the metabolites in rows and biological replicates in columns (green: control; red: experimental). The red and blue gradients represent expression levels above/below the global mean, respectively.

### 2.7. Metabolic Pathway Enrichment Analysis

KEGG pathway analysis identified 38 significantly enriched pathways (*p* <0.05), with the top 10 illustrated in Figure 5. These include tryptophan metabolism, arginine/proline metabolism, and ABC transporters. Bubble plots display the enrichment factors (*x*-axis), metabolite counts (point size), and −log10 (*p*-value) (color gradient).

### 2.8. Tryptophan Metabolic Pathway Analysis

Tryptophan metabolism emerged as the most significantly altered pathway. Eight metabolites were upregulated in HE rats (Figure 6), including L-tryptophan (L-Trp), L-kynurenine (L-KYN), quinolinic acid (QUIN), 5-HIAA, N-acetylserotonin (NAS), indole, indole-3-pyruvate (I3P), and 3-indoleacetonitrile (IAN). Clustering heatmaps (Figure 6A) and boxplots (Figure 6B) confirmed systemic activation of tryptophan metabolism. The observed dysregulation of the tryptophan pathway aligns with canonical metabolism (KEGG map ID: hsa00380), with kynurenine and quinolinic acid emerging as dominant mediators in both human and rat cohorts. The concordant upregulation of tryptophan-derived metabolites (e.g., kynurenine, quinolinic acid) in both human MR analysis and rat models supports the biological robustness of these findings, despite the reliance on relative quantification in rats.

## 3. Discussion

This study combined MR analysis of serum metabolites in hypoglycemic encephalopathy (HE) patients with untargeted metabolomics profiling in insulin overdose-induced HE rat models. The results revealed that the major differential metabolic pathways in the serum of HE primarily involve tryptophan and glutamate metabolism. In forensic investigations, critical postmortem artifacts often compromise the reliability of traditional biomarkers. Our study illustrates how MR can overcome these challenges. While conventional biochemical markers are vulnerable to pre-analytical degradation—especially due to the instability of glutamate in compromised specimens—the tryptophan metabolites identified through MR maintain their diagnostic reliability even in difficult forensic contexts. This robustness stems from our innovative dual-methodological approach: first, we used genetic instrumental variables that inherently avoid postmortem-specific confounders, and second, we provided experimental validation that shows consistent pathway dysregulation between human and mammalian models. To our knowledge, this research is the first to combine causal inference methods with forensic metabolomics, positioning MR-curated metabolites as a groundbreaking approach to identifying postmortem hypoglycemic encephalopathy. Our dual methodological approach provides converging evidence for the causal hierarchy between HE and metabolic dysregulation. By specifically selecting genetic instruments associated with HE susceptibility rather than metabolite-related variants, the MR analysis establishes HE as the upstream driver of tryptophan/glutamate pathway alterations. This directional relationship (HE → metabolites) was further validated experimentally. The consistency between genetic causality estimates and time-resolved metabolic remodeling underscores the fact that HE-initiated metabolic rewiring represents downstream consequences rather than upstream contributors.

Tryptophan is an essential amino acid obtained exclusively through dietary intake, and it serves as the biochemical precursor for critical metabolites such as serotonin (5-HT), melatonin, and niacin [16]. Tryptophan and its metabolites play significant roles in various physiological and pathological processes, including neuropsychiatric disorders (e.g., depression, schizophrenia) [17,18], cancer [19], inflammatory bowel disease [20], and cardiovascular diseases [21]. In the present study, differential metabolites associated with tryptophan metabolism were found in both HE patients and HE rats, including tryptophan, kynurenine, quinolinic acid, 5-hydroxyindoleacetic acid (5-HIAA), N-acetylserotonin (NAS), indole, indolelactate, indole-3-pyruvate, and 3-indoleacetonitrile. Tryptophan is primarily metabolized through three pathways, namely the kynurenine pathway, serotonin pathway, and indole pathway [22].

Kynurenine pathway (KP):

The KP accounts for 95% of tryptophan metabolism in humans [23]. In this study, elevated levels of kynurenine were found in both HE patients and HE rats, whereas increased quinolinic acid levels were observed in HE rats only. Kynurenine, an intermediate metabolite of the KP, is generated from tryptophan via the action of tryptophan 2,3-dioxygenase (TDO) or indoleamine 2,3-dioxygenase (IDO), forming N-formylkynurenine, which is subsequently converted to kynurenine. Kynurenine is further metabolized into kynurenic acid, picolinic acid, xanthurenic acid, and quinolinic acid [24]. The elevation of kynurenine suggests activation of the KP. Quinolinic acid, a potent agonist of the N-methyl-D-aspartate (NMDA) receptor in the brain [25], has been shown to induce seizures in mice following intracerebroventricular injection [26]. Heyes et al. reported a 6.5-fold increase in plasma quinolinic acid levels during insulin-induced hypoglycemia, with a 2–3-fold increase in the brain [27]. Moreover, the KP is the sole de novo synthesis pathway for nicotinamide adenine dinucleotide (NAD+). Quinolinic acid phosphoribosyltransferase (QPRT) is a key rate-limiting enzyme in NAD+ synthesis that facilitates the conversion of quinolinic acid into nicotinamide mononucleotide (NMN), which is further adenylated to NAD+. Chini et al. proposed that an elevated quinolinic acid-to-tryptophan (Q/T) ratio may indicate reduced QPRT activity [28]. Wang et al. demonstrated that NMN administration alleviates hippocampal damage in severely hypoglycemic rats, suggesting that impaired NAD+ synthesis due to QPRT deficiency may play a role in hypoglycemic brain injury [29]. In our study, the elevated Q/T ratio in HE rats implies diminished QPRT activity. Moreover, concurrent increases in nicotinamide, another NAD+ precursor, further indicate disrupted NAD+ synthesis. These findings suggest that KP activation is not merely a consequence of HE, as it may also drive cerebral damage.

Serotonin pathway:

The serotonin pathway accounts for 1–2% of total tryptophan metabolism. Tryptophan is converted to 5-hydroxytryptophan by tryptophan hydroxylase (TPH), which is further metabolized to serotonin (5-HT) and ultimately into melatonin in the pineal gland [24]. 5-HIAA is the primary breakdown product of 5-HT and is generated in the liver, serving as a surrogate marker for 5-HT levels in 24 h urine assays [30]. Notably, elevated serum 5-HIAA may reflect increased 5-HT turnover. NAS, an intermediate in melatonin synthesis from 5-HT, exhibits antioxidant, anti-apoptotic, and anti-autophagic properties [31]. The increases in 5-HIAA and NAS levels in our study indicate activation of the serotonin pathway.

Indole pathway:

The indole pathway, predominantly mediated by gut microbiota, converts tryptophan into various indole derivatives [24]. In this study, elevated levels of serum differential metabolites were observed, including indole, indolelactate, indole-3-pyruvate, and 3-indoleacetonitrile, which are all products of tryptophan metabolism. These findings are consistent with overall pathway activation. Indole derivatives regulate intestinal permeability, inflammation, and host immunity [32,33,34], suggesting that indole pathway activation may reflect heightened inflammatory responses.

Glutamate excitotoxicity: hypoglycemia-related neuronal death involves a cascade of processes rather than direct energy failure. Excitotoxic mechanisms, particularly glutamate toxicity, are implicated in hypoglycemic brain injury [35]. Notably, hypoglycemia elevates extracellular glutamate levels, correlating with neuronal death in rats [36]. In addition, pretreatment with glutamate receptor antagonists was found to reduce hypoglycemia-induced neuronal loss [37]. Cardoso et al. reported increased plasma levels of aspartate, glutamate, glutamine, and taurine in streptozotocin-induced diabetic rats following insulin overdose, whereas decreased GABA levels were observed [38]. Conversely, Gundersen et al. revealed that the levels of aspartate were elevated and glutamate/glutamine was reduced in the hippocampus and striatum of hypoglycemic rats [39]. In the present study, decreased serum glutamate levels were found in HE patients; in contrast, both aspartate and glutamate levels declined in HE rats, with concomitant increases in GABA and taurine. These findings may reflect a shift from excitatory to inhibitory neurotransmission during late-stage HE. However, serum neurotransmitter levels do not directly mirror cerebral concentrations, necessitating further research into the relationship between central and peripheral glutamate dynamics in hypoglycemic brain injury.

This study not only elucidates the activation of tryptophan and glutamate pathways in HE patients and rat models, but also identifies novel biomarkers for early diagnosis. The metabolic alterations provide insights into the effects of hypoglycemia on cerebral function and potential links to neuropsychiatric disorders such as depression and anxiety. Furthermore, the identification of HE-associated metabolites may facilitate personalized therapeutic strategies, enabling clinicians to tailor interventions based on individual metabolic profiles. Key pathway components, including kynurenine and quinolinic acid, represent promising targets for drug development. This study integrates foundational metabolomics with clinical data to establish a critical theoretical and practical framework for advancing HE management.

There are some limitations to our study. 1. Our study identified diagnostic biomarkers for hypoglycemic encephalopathy but did not explore their association with disease severity. Future studies incorporating longitudinal clinical data (e.g., Glasgow Coma Scale scores, neuroimaging findings) and quantitative neuropathological assessments in experimental models are needed to establish severity-related biomarkers. 2. While our findings are derived from male rats, future studies should investigate whether the observed tryptophan–glutamate axis alterations are conserved in females, given the known sex disparities in antioxidant defense mechanisms. 3. While lipids were part of our MR screening, their limited presence among the causal metabolites and their vulnerability to postmortem confounding led us to concentrate on the tryptophan–glutamate axis. 4. While our rat metabolomics data are based on relative quantification, the cross-species validation with human MR results (which are immune to batch effects) mitigates concerns of measurement bias. Future studies will prioritize absolute quantitation of the identified biomarkers (e.g., the kynurenine-to-glutamate ratio) for forensic applications. 5. While most metabolites were endogenous, trace contaminants (e.g., phenazone) may reflect environmental exposure in human cohorts. Future studies will employ contaminant-specific SPE cartridges during sample preparation.

## 4. Materials and Methods

The main technical workflow is illustrated in Figure 7.

### 4.1. GWAS Data from Human Plasma Metabolites

This research aimed to summarize the statistics related to plasma metabolites derived from a comprehensive GWAS that examined 850 distinct metabolites. The study included data from 8299 participants of European descent drawn from the Canadian Longitudinal Study on Aging (CLSA) [40]. Among the 850 identified metabolites, recognized characteristics were distributed across eight different biological pathways, including lipids, amino acids, xenobiotics, nucleotides, cofactors and vitamins, carbohydrates, peptides, and energy pathways. Additionally, the summary statistics for 1400 plasma metabolites can be found on the GWAS Catalog platform at https://www.ebi.ac.uk/gwas/publications/36635386 (accessed on 11 May 2024). The GWAS IDs of all plasma metabolites are provided in Appendix A. The original data can be downloaded by entering the IDs into the search box of the GWAS Catalog platform. All original GWAS studies were approved by an ethics committee, and written informed consent was obtained from each participant before data collection.

### 4.2. GWAS Data for Hypoglycemic Encephalopathy

The GWAS summary statistics for hypoglycemic encephalopathy were retrieved from the United Kingdom Biobank, which includes 519 cases of hypoglycemic encephalopathy and 455,829 controls of European ancestry. Blood collection was initiated within 2 h after hospital admission, with a median sampling time of 3.2 h (IQR: 2.8–4.1) post overdose event. The distributions of the severity of hypoglycemic encephalopathy, gender, age, and other information are included in Appendix A. A generalized linear mixed model (GLMM) approach, known as fast GWA-GLMM, was utilized with appropriate covariate adjustments [41].

### 4.3. Selection of Instrumental Variables

Mendelian randomization (MR) is an essential technique for establishing causal connections between traits. It uses genetic variation as an instrumental variable (IV) in GWAS data to identify the causal relationship between exposure and outcomes. In this study, hypoglycemic encephalopathy was treated as the exposure variable, and blood metabolites were set as outcomes. The selected IVs must meet three primary assumptions [42]: (1) the genetic variation is connected to the exposure; (2) the relationship between genetic variation and exposure outcomes is not affected by confounding factors; (3) genetic variation influences the outcome solely through the exposure factors. To adhere to these assumptions, SNPs were selected as IVs by applying a significance threshold of *p* < 5 × 10^−6^ to discard insignificant and highly correlated SNPs, maintaining independence [43]. To ensure independence and minimize linkage disequilibrium (LD), thresholds of R^2^ < 0.001 and a distance of 10,000 kb were employed. The strength of the identified SNPs as IVs was evaluated by calculating the F-statistic for each metabolite, with an F > 10 threshold indicating robust IVs for subsequent MR analysis [44]. The selection of SNPs for hypoglycemic encephalopathy is detailed in Appendix A.

### 4.4. Statistical Methods

The inverse variance weighted (IVW) method has been widely applied in MR studies and is recognized for its robustness. This technique operates under both fixed and random effect models to reduce bias arising from heterogeneity. All SNPs included in the IVW analysis must adhere to the three IV selection hypotheses, especially the exclusivity hypothesis, which states that genetic variation should influence outcomes exclusively through the studied exposure factors. Despite efforts to exclude confounding SNPs, horizontal pleiotropy may affect causal effect estimations. Consequently, MR Egger regression and weighted median estimator (WME) methods were employed to evaluate the stability of our results. The MR Egger regression adapts the IVW approach to assess horizontal pleiotropy and correct for biases. However, the MR Egger regression provides limited reliability for causal estimation. Some researchers suggest using MR Egger primarily as a sensitivity analysis to check for any violations of IV assumptions rather than as a replacement for the IVW method. Conversely, the WME method can produce consistent results, particularly when certain genetic variations do not function effectively as IVs. Positive results were determined when the estimates from all three methods agreed and the IVW results were significant (PIVW < 0.05) [45].

Sensitivity analysis included tests for heterogeneity and multiplicity. Differences in the included studies, such as varying gene annotations, analytical platforms, or criteria for inclusion/exclusion, may result in heterogeneity. In this study, heterogeneity in the IVW and MR Egger methods was assessed using Cochran’s Q-test, which indicated a *p*-value greater than 0.05, indicating no significant impact on the outcomes. As the intercept term in MR Egger regression approaches zero, the degree of horizontal pleiotropy diminishes. A *p*-value greater than 0.05 in the horizontal pleiotropy assessment suggests the absence of such pleiotropy. Specifically, the MR-PRESSO outlier test identifies SNPs that could skew the overall results. In addition, leave-one-out (LOO) analysis methodically sequentially excludes SNPs to evaluate their influence, and the stability of the outcomes are presented in a forest plot. The MR Steiger test further validates the direction of causality.

### 4.5. Experimental Animals and Grouping

Healthy adult male Sprague-Dawley (SD) rats (weight: 275–325 g, age: 8–10 weeks) were provided by Bennt Biological Technology Co., Ltd, Wuhan, China. The rats were randomly assigned to the experimental and control groups (*n* = 10 per group). Male rats were selected to control for sex-dependent differences in insulin sensitivity and cerebral glucose metabolism, as reported in prior hypoglycemia models [46]. The animals were housed in a controlled environment (temperature: 20–25 °C, humidity: 40–60%) with a 12 h light/dark cycle. Each cage contained 3–4 rats, which were allowed free access to food and water. All experimental procedures complied with the guidelines approved by the Animal Ethics Committee.

### 4.6. Establishment of Insulin Overdose-Induced Hypoglycemic Encephalopathy Rat Model

After fasting for 16–20 h, the body weight and baseline blood glucose (via tail vein sampling) levels of the rats were recorded. Rats in the experimental group received an intraperitoneal (IP) injection of 20 IU/kg protamine human insulin. Blood glucose was measured every 30 min, and behavioral changes were recorded. When blood glucose dropped below 2.5 mmol/L, the rats were placed in an anesthesia induction chamber. Subsequently, isoflurane was administered at 2.0% for 3 min, then increased to 4.0–5.0% for 5 min until full anesthesia was achieved.

Tracheal intubation was performed with ventilator parameters set to 70 breaths/min and a tidal volume of 6–8 mL. Rats were secured in a supine position, and electrodes coated with petroleum jelly were placed over the pre-auricular, occipital, and nuchal regions. Hypoglycemic encephalopathy was confirmed using continuous electroencephalogram (EEG) monitoring (flat EEG waves persisting for ≥1 h) [12]. The rats in the control group received a saline injection of equivalent volume, followed by identical procedures.

### 4.7. Serum Sample Collection and Pretreatment

After EEG monitoring, blood was collected by cardiac puncture from the right ventricle using a vacuum tube (4–5 mL). Samples were clotted at room temperature for 30 min, then centrifuged (4 °C, 2000 rpm, 10 min), and serum aliquots were stored in liquid nitrogen for 24 h before transfer to a −80 °C freezer.

For metabolomic analysis, serum samples were thawed at −20 °C for 1–2 days and then at 4 °C. Aliquots (100 μL) were mixed with 700 μL ice-cold extraction solvent (methanol:acetonitrile:water = 4:2:1, with internal standard), vortexed for 1 min, incubated at −20 °C for 2 h, and centrifuged (4 °C, 25,000 rpm, 15 min). Supernatants were dried using nitrogen gas, reconstituted in 180 μL methanol:water (1:1), centrifuged again, and filtered for LC-MS analysis.

### 4.8. UPLC-MS/MS Analysis

Chromatography was performed using a BEH C18 column (1.7 μm, 2.1 × 100 mm; Waters, Milford, MA, USA). Mobile phase (positive ion mode): 0.1% formic acid in water (A) and 0.1% formic acid in methanol (B). Mobile phase (negative ion mode): 10 mM ammonium formate in water (A) and 10 mM ammonium formate in 95% methanol (B). Column temperature 45 °C; injection volume 5 μL. The gradient elution parameters are detailed in Appendix B.

Mass spectrometry conditions: Data acquisition was performed using a Q Exactive HF mass spectrometer (Thermo Fisher Scientific, Waltham, MA, USA) in full-scan MS/MS mode. Detailed parameters are provided in Appendix C.

### 4.9. Data Processing and Analysis

Raw MS data were processed using Compound Discoverer 3.3 (retention time, *m*/*z* tolerance: ±5 ppm for precursors, ±10 ppm for fragments). Peaks were aligned and annotated using the BMDB, mzCloud, and ChemSpider databases. Metabolite identification required the following: (a) m/z error <5 ppm; (b) retention time (RT) deviation <0.3 min vs. standards; (c) MS/MS dot-product ≥0.8 against mzCloud/HMDB. Level 1–5 annotations followed the guidelines of Schymanski et al. (2014) [47]. Only Level 1–2 identifications were retained for downstream analysis. Suspected contaminants (e.g., drugs, industrial chemicals) were excluded unless verified by reference standards.

Data were imported into MetaboAnalyst 5.0 for preprocessing, which involved several key steps to ensure the integrity and reliability of the analysis. First, normalization was performed using probabilistic quotient normalization (PQN), with quality control (QC) samples serving as the reference. This step helps to adjust for systematic biases in the data. Next, the ComBat algorithm was employed for batch correction, effectively removing any technical batch effects that could confound the results. Following this, data filtering was conducted to eliminate metabolites with a relative standard deviation (RSD) greater than 30% in the QC samples, ensuring that only reliable data were retained for further analysis. Finally, unit variance scaling, also known as auto-scaling, was applied to normalize the variance of the features, allowing for a more accurate comparison across the dataset. Subsequently, multivariate analysis (PCA, OPLS-DA) and pathway enrichment (KEGG, HMDB) were performed. Differential metabolites were identified with VIP ≥ 1, Q-value < 0.05 (FDR-corrected), and fold change ≥1.5 or ≤0.67.

Statistical analysis (*t*-tests, GraphPad Prism 9.0.0) and visualization were performed. Data were expressed as the mean ± SEM. In this study, *p* < 0.05 is considered statistically significant.

## 5. Conclusions

Significant differences in serum metabolic profiles were observed between HE patients, insulin overdose-induced HE rats, and controls. Tryptophan metabolism was upregulated, accompanied by decreased serum glutamate levels. Activation of the kynurenine pathway within tryptophan metabolism may serve both as a consequence and a causative factor in hypoglycemic brain injury, though the precise mechanisms require further investigation. The metabolic changes in the tryptophan pathway provide new insights into the impact of hypoglycemia on brain function. The metabolites associated with tryptophan metabolism hold potential value for HE diagnosis and the forensic identification of fatalities caused by insulin overdose.

## Figures and Tables

**Figure 1 ijms-26-04152-f001:**
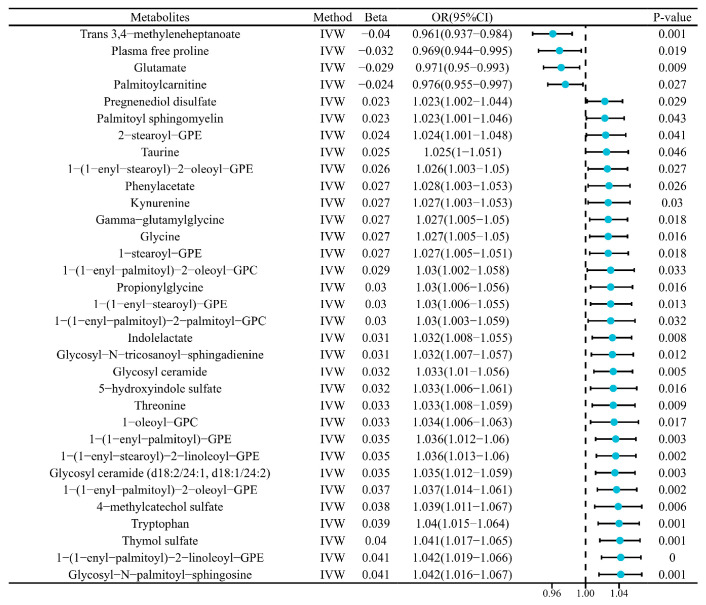
33 metabolites significantly associated with hypoglycemic encephalopathy. Solid blue dots represent odds ratios (ORs), horizontal lines indicate 95% confidence intervals, and dashed vertical lines mark null effects (OR = 1). The magnitude of effect size is encoded in dot position relative to the vertical null line, with horizontal line length reflecting estimation precision.

**Figure 2 ijms-26-04152-f002:**
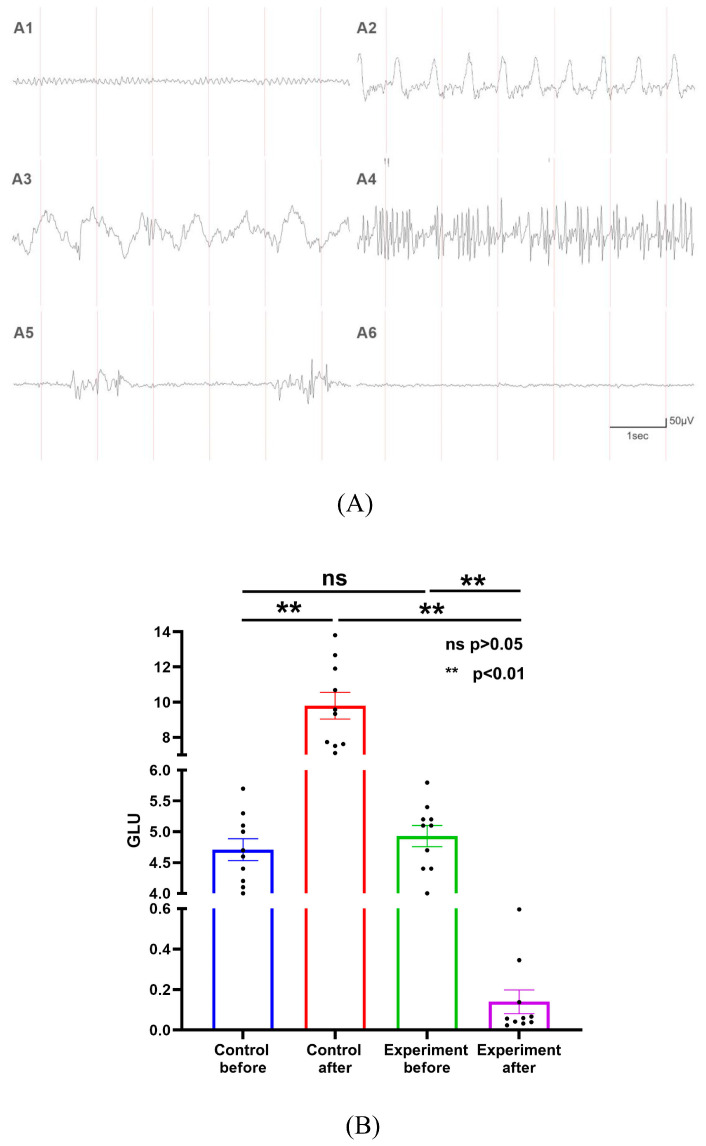
(**A**) EEG changes during insulin overdose-induced hypoglycemic encephalopathy. (**A1**) Initial EEG monitoring shows stable α waves. (**A2**) As blood glucose levels decrease, EEG displays rhythmic δ waves. (**A3**) EEG further changes to high-amplitude δ waves at 1.5–2 Hz. (**A4**) Some rats exhibit polyspike waves in EEG during epileptic seizures. (**A5**) Some rats show burst suppression phenomenon before EEG becomes isoelectric. (**A6**) Isopotential period of EEG. (**B**) Blood glucose profiles in experimental and control rats.

**Figure 3 ijms-26-04152-f003:**
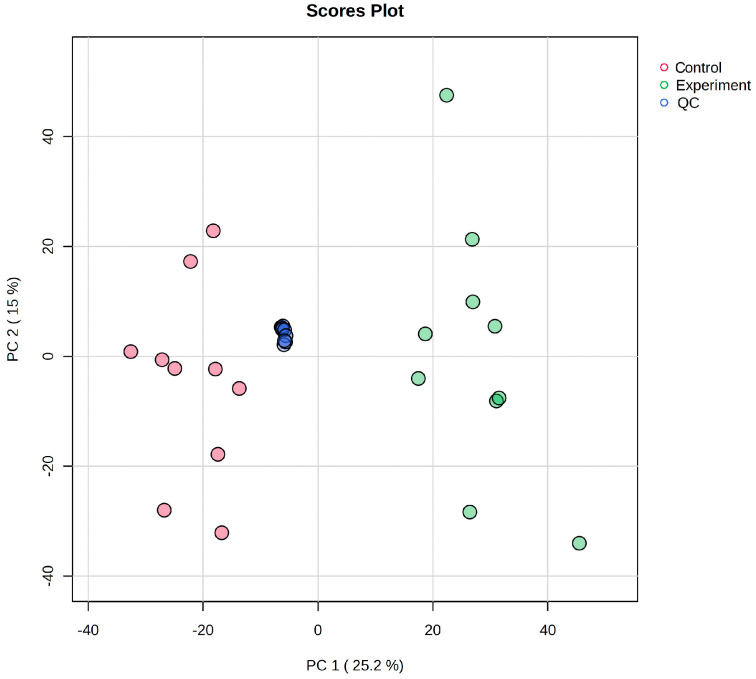
PCA score plot of all samples.

**Figure 4 ijms-26-04152-f004:**
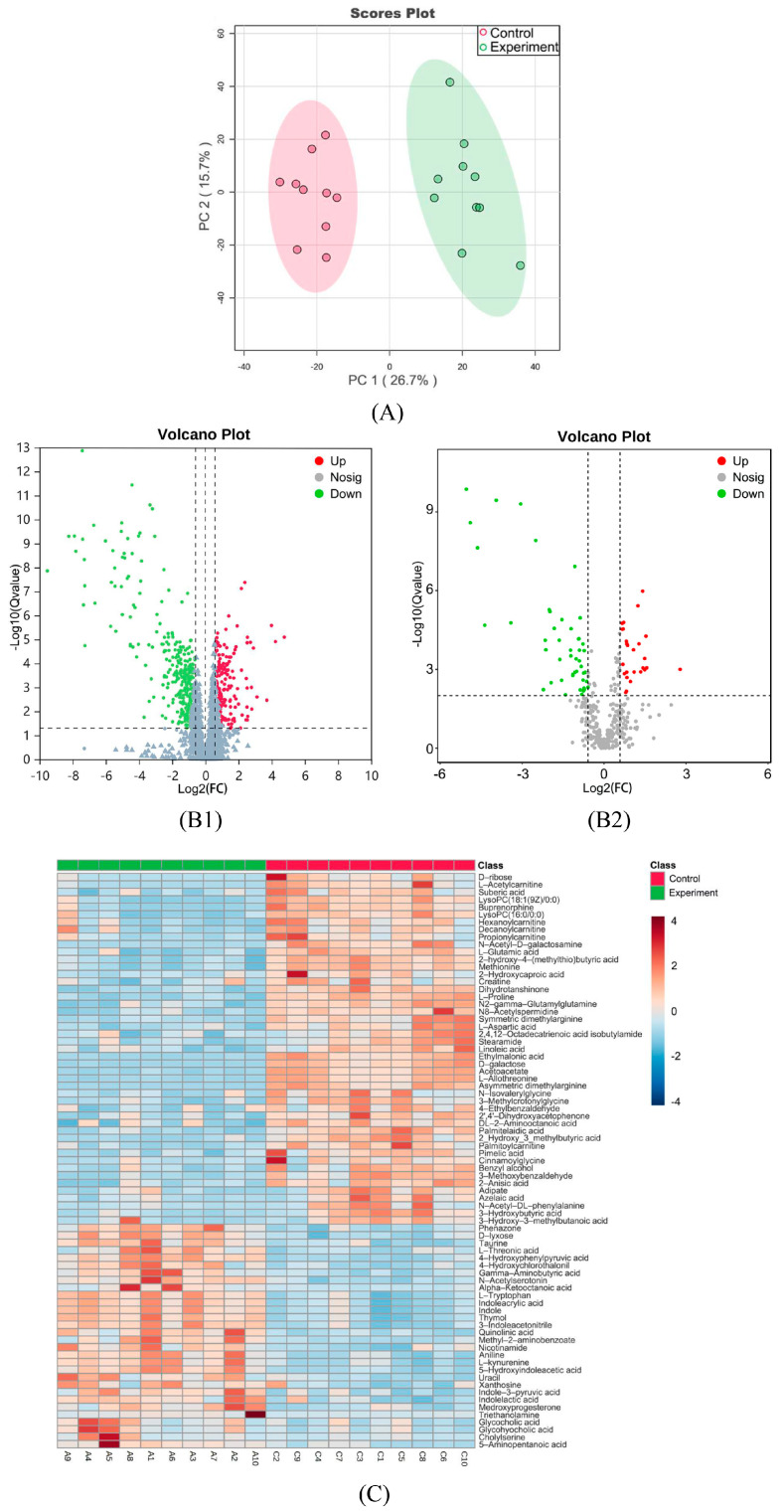
(**A**) PCA score plot. (**B**) Volcano plots: (**B1**) all metabolites; (**B2**) endogenous metabolites. The dashed line represents the set screening criteria, with FC ≥1.5 or ≤0.67. (**C**) Heatmap of endogenous differential metabolite expression.

**Figure 5 ijms-26-04152-f005:**
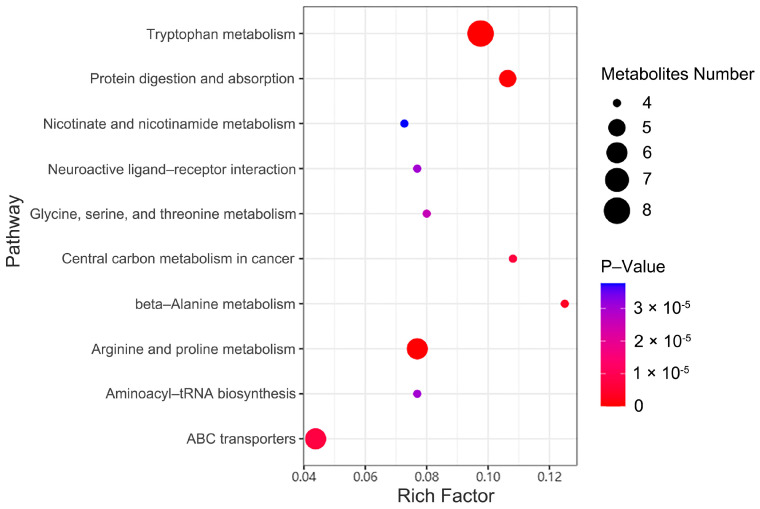
Bubble plot of enriched metabolic pathways.

**Figure 6 ijms-26-04152-f006:**
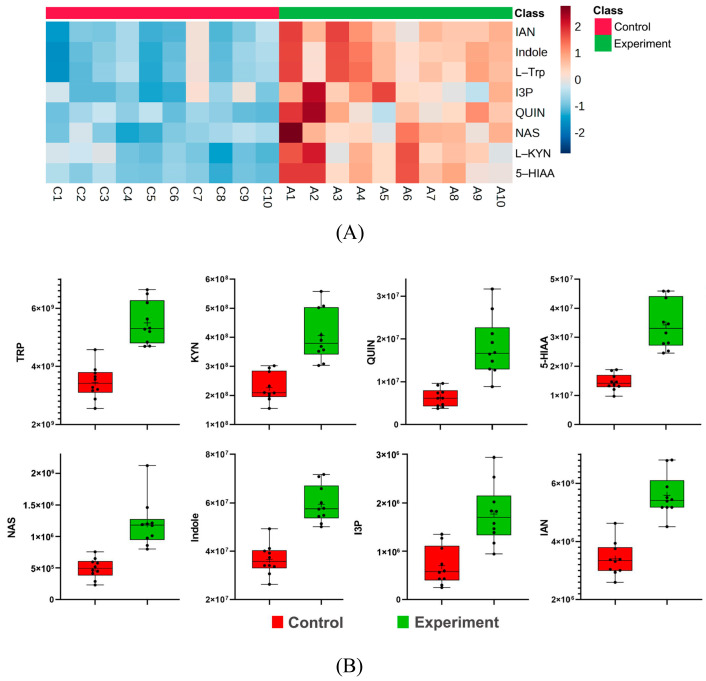
(**A**) Heatmap of tryptophan pathway metabolites. (**B**) Boxplots of tryptophan pathway metabolites.

**Figure 7 ijms-26-04152-f007:**
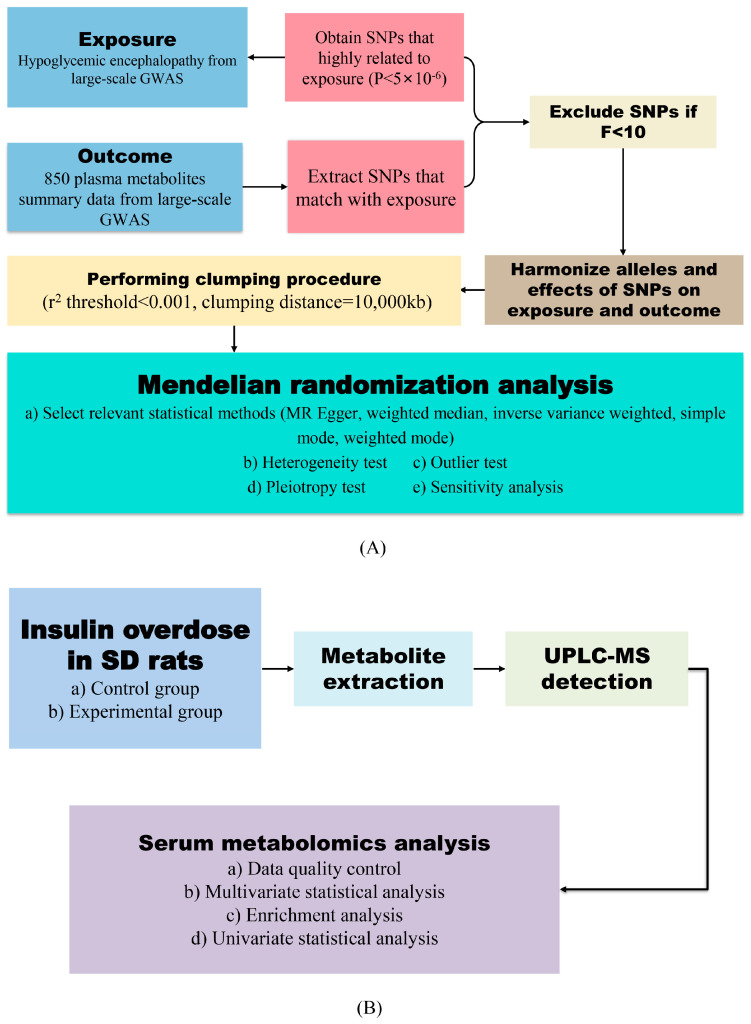
Schematic diagram of experimental workflow: (**A**) Design of causal association study involving hypoglycemic encephalopathy patients and serum metabolites based on two-sample Mendelian randomization analysis. (**B**) Serum metabolomics analysis in rat model of insulin overdose-induced hypoglycemic encephalopathy. Solid arrows indicate primary analytical workflows.

## Data Availability

All data produced or examined in this study can be found in the published article and its Appendix A.

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
