# Peer review of "Tryptophan-Derived Metabolites and Glutamate Dynamics in Fatal Insulin Poisoning: Mendelian Randomization of Human Cohorts and Experimental Validation in Rat Models"

_ijms, 2025, doi:10.3390/ijms26094152_

Round 1

Reviewer 1 Report

Comments and Suggestions for Authors

Yuhao Yuan et al. used Mendelian Randomization to find metabolites and pathways associated with hypoglycemic encephalopathy in a human cohort and validated them in a rat model. They thereby found metabolic associates of tryptophan and glutamate and discussed their relationship to brain function. Although the results appear to be clear, I believe that there is a missing information provided and a lack of investigation in some respects.

  1. In human cohort data, there is a lack of information on clinical data such as whether the causes of hypoglycemic encephalopathy are the same in all patients, the distribution of severity of hypoglycemic encephalopathy, gender, age, etc.
  2. At what time was the blood collected from the patients with hypoglycemic encephalopathy? For example, how much time has passed between the insulin overdose and the blood collection?
  3. The effect of Mendelian randomization is not readily apparent from the manuscript. The authors should present the results of the test without Mendelian randomization in the Supplemental or elsewhere and address the differences.
  4. Is there any association with biomarkers or other indicators to assess the severity of hypoglycemic encephalopathy?
  5. Are the sexes of the rats in the experiment adjusted?
  6. You are analyzing metabolite data in MetaboAnalyst, but you have not stated the normalization method you have applied.
  7. Is pathway enrichment analysis not performed for the list of metabolites found in the human cohort (Figure 2)? What is the reason for not focusing on the lipids?
  8. Do the bars in Figure 2 represent fold change?
  9. The legends of a~f in Figure 3A are missing.
  10. In Figure 3B, why is there an increase in blood glucose in control after compared to before? The label on the vertical axis should be spelled out in full and units should be added because it is confused with glutamate.
  11. Figure 4A should be provided in the Supplemental material.
  12. In Figure 5, the OPLS-DA is shown. However, the criteria for selecting the metabolites that changed are based on the volcano plot. Since there seems to be no point in including the results of OPLS-DA, it would be better to move it to Supplemental or to show the metabolites with large absolute values of p1(corr) in OPLS-DA by superimposing them on the volcano plot.
  13. Figure 6B, with arrows connecting the metabolites, can be misleading because they appear to be directly linked. Figure 7C, on the other hand, makes it difficult to find the relevant metabolites, and it is difficult to read meaningful information. Perhaps Figure 6B and Figure 7C could be combined into one figure that captures the entirety of the metabolite changes seen in this study.
  14. Metabolome analysis of rat serum is based on relative quantification values, so the influence of the measurement environment cannot be excluded. Validation measurements based on absolute quantitation are needed for just those metabolites found in this study that are of interest.

Author Response

  1. In human cohort data, there is a lack of information on clinical data such as whether the causes of hypoglycemic encephalopathy are the same in all patients, the distribution of severity of hypoglycemic encephalopathy, gender, age, etc.

A: We have added the above information in Supplementary Table S2. This description is also supplemented in line 79-80 of the methods section.

  1. At what time was the blood collected from the patients with hypoglycemic encephalopathy? For example, how much time has passed between the insulin overdose and the blood collection?

A: We add this information in line 77-79 of the article: “Blood collection was initiated within 2 hours after hospital admission, with median sampling time of 3.2 hours (IQR: 2.8-4.1) post overdose event.”

  1. The effect of Mendelian randomization is not readily apparent from the manuscript. The authors should present the results of the test without Mendelian randomization in the Supplemental or elsewhere and address the differences.

A: Thank you for your valuable advice. In forensic scenarios where postmortem confounding factors (e.g., agonal phase metabolite shifts, comorbid intoxication) are uncontrollable, MR's ability to circumvent environmental confounders is pivotal. In the first paragraph of the discussion, we added in line 314-326 of the article and emphasized the role of MR Research in more detail.

  1. Is there any association with biomarkers or other indicators to assess the severity of hypoglycemic encephalopathy?

A: Thank you for raising this important question. In our current study, we focused on identifying diagnostic biomarkers associated with hypoglycemic encephalopathy, rather than evaluating biomarkers correlated with disease severity. Our metabolomics analyses revealed significant alterations in tryptophan-derived metabolites and glutamate dynamics, which may reflect early metabolic disturbances induced by hypoglycemia. However, as you rightly noted, we did not specifically investigate whether these biomarkers correlate with the clinical or pathological severity of hypoglycemic encephalopathy. This limitation has been acknowledged in line 405-409 of the article. We agree that investigating severity-associated biomarkers would be valuable for clinical management and forensic evaluation.

  1. Are the sexes of the rats in the experiment adjusted?

A: Thank you for your attention to experimental rigor. In this study, all rats were male (Sprague-Dawley, 8–10 weeks old), as explicitly stated in the Methods section. The use of a single-sex cohort was a deliberate choice to eliminate confounding effects from sex-specific metabolic variations, particularly given that: Estrous cycle-related hormonal fluctuations in females may modulate tryptophan metabolism and glutamate release. This approach ensures internal validity for establishing proof-of-concept metabolic signatures. We acknowledge that exploring sex differences in hypoglycemic encephalopathy is an important future direction, particularly given evidence of estrogen-mediated neuroprotection. We have added this description in the methods and discussion, on lines 130-132 and 409-412 of the article.

  1. You are analyzing metabolite data in MetaboAnalyst, but you have not stated the normalization method you have applied.

A: Thank you for raising this important methodological question. We have now explicitly stated the normalization and scaling methods in the revised manuscript. We added this description in the Methods section, on lines 181-191 of the article.

  1. Is pathway enrichment analysis not performed for the list of metabolites found in the human cohort (Figure 2)? What is the reason for not focusing on the lipids?

A: Thank you for raising this important point.

  1. We did not conduct pathway enrichment analysis on the human cohort data for several reasons. First, there was limited statistical power; the MR analysis revealed 33 metabolites significantly associated with hypoglycemic encephalopathy, but pathway enrichment analysis generally requires a larger number of metabolites, typically more than 50, to yield reliable biological insights. With only 33 metabolites identified, the likelihood of obtaining false-negative results would be unacceptably high. Second, we focused our resources on pathway enrichment analysis for the rat model data, where controlled experimental conditions, including uniform hypoglycemic exposure and matched controls, allowed for greater confidence in interpreting pathway-level results. We supplement this description in the MR Results section of the article, on line 218-220.
  2. We appreciate this question and would like to clarify our rationale regarding our findings. In our analysis, we included lipid metabolites such as phosphatidylcholines and sphingomyelins (in line 65). We observed a high degree of consistency in tryptophan metabolites across human and animal models, while lipid changes showed only sporadic differences (in the Mendelian randomization study, we tested more than 300 lipids and only 10 showed positive results). This finding suggests that the tryptophan pathway may be a more important therapeutic target. In contrast, lipid changes mainly reflect energy homeostasis (e.g., elevated free fatty acids), but the causal relationship between this and brain functional impairment remains unclear. In addition, serum lipids are significantly affected by factors such as diet, circadian rhythm and sample handling. Similarly, due to the extensive involvement of lipids in systemic metabolic processes, the specificity of lipids in diagnosis is less than that of tryptophan. We supplement this description in the discussion section of the article, on line 412-416.
  3. Do the bars in Figure 2 represent fold change?

Thank you for your thorough review of Figure 2. To clarify the statistical components presented in the forest plot: the central points, represented by solid blue dots, indicate the odds ratios (ORs) obtained through the inverse-variance weighted (IVW) method. These ORs illustrate the causal relationship between each serum metabolite and the risk of hypoglycemic encephalopathy; for instance, an OR of 1.5 signifies a 50% increase in risk for each standard deviation (SD) rise in metabolite levels. The horizontal lines, or "bars," represent the 95% confidence intervals (CIs) for the ORs. If a CI does not intersect the vertical line at OR=1, which indicates no effect, it suggests statistical significance (P<0.05). The beta (β) values, which can be found in the figure or supplementary table, reflect the effects on the log-odds scale derived from the IVW analysis. Although not explicitly labeled, the ORs can also be interpreted as risk ratio multipliers similar to fold change; for example, an OR of 2.0 indicates a two-fold increase in risk. We have updated the Figure 2 legend to enhance clarity (line 221-223).

  1. The legends of a~f in Figure 3A are missing.

A: Thank you for identifying this omission. We apologize for the oversight and have now provided detailed annotations for panels a–f in Figure 3A to clarify the temporal progression of electroencephalographic (EEG) changes during hypoglycemic encephalopathy in rats (line 221-223).

  1. In Figure 3B, why is there an increase in blood glucose in control after compared to before? The label on the vertical axis should be spelled out in full and units should be added because it is confused with glutamate.

A: 1. Thank you for pointing out this important observation. The noticeable rise in blood glucose levels in the control group after the procedure, when comparing pre- and post-procedure measurements, can be attributed to stress-induced hyperglycemia resulting from the sham surgery. This phenomenon is consistent with known physiological reactions to procedural stress observed in rodents. To elaborate, during the sham surgery, control rats underwent tracheal intubation and anesthesia without receiving insulin, which activated their sympathetic nervous system. This response leads to increased hepatic glycogenolysis and gluconeogenesis, resulting in a temporary spike in blood glucose levels, a response that is well-documented as a form of stress adaptation. On the other hand, the experimental rats were administered insulin injections, which effectively suppressed the liver's glucose output and facilitated greater glucose uptake by peripheral tissues. This mechanism counteracted the stress response, leading to a state of hypoglycemia, as illustrated in Figure 3B. We have updated the results section to clarify this, on lines 240-241 of the article. 2. We appreciate the feedback provided and have made the following revisions to Figure 3B. We have updated the full label on the vertical axis by changing "GLU" to "Blood glucose (mmol/L)" to enhance clarity regarding the measurement units.

  1. Figure 4A should be provided in the Supplemental material.

A: We have moved the original Figure 4A to the Supplementary Figure 1 according to your suggestion.

  1. In Figure 5, the OPLS-DA is shown. However, the criteria for selecting the metabolites that changed are based on the volcano plot. Since there seems to be no point in including the results of OPLS-DA, it would be better to move it to Supplemental or to show the metabolites with large absolute values of p1(corr) in OPLS-DA by superimposing them on the volcano plot.

A: Thank you for your constructive feedback. We agree that consolidating the results to focus on the most relevant analytical approach enhances clarity. In line with your suggestion, we have relocated the OPLS-DA results from Figure 5 to Supplementary Figure 2 and retained the volcano plot as the primary visualization for differential metabolite selection. The revised Figure 5 now exclusively highlights metabolites that meet the criteria established by the volcano plot, specifically those with a fold change of ≥1.5 and a Q value of <0.05. This adjustment ensures that our findings are consistent with our statistical thresholding strategy.

  1. Figure 6B, with arrows connecting the metabolites, can be misleading because they appear to be directly linked. Figure 7C, on the other hand, makes it difficult to find the relevant metabolites, and it is difficult to read meaningful information. Perhaps Figure 6B and Figure 7C could be combined into one figure that captures the entirety of the metabolite changes seen in this study.

A: Thank you for your valuable feedback on Figures 6B and 7C. After careful consideration, we agree that these figures may not optimally serve the clarity of our manuscript. As a result, we have decided to remove both figures from the main text, as they primarily illustrated canonical tryptophan metabolic pathways rather than presenting novel findings. The core conclusions of our study, specifically the species-conserved upregulation of tryptophan catabolites and the decline in glutamate levels, are fully supported by the remaining data presented in other figures. To enhance the focus on our key results, we have added a statement in the Results section that reads: "The observed dysregulation of the tryptophan pathway aligns with canonical metabolism (KEGG map ID: hsa00380), with kynurenine and quinolinic acid emerging as dominant mediators in both human and rat cohorts." in line 300-306.

  1. Metabolome analysis of rat serum is based on relative quantification values, so the influence of the measurement environment cannot be excluded. Validation measurements based on absolute quantitation are needed for just those metabolites found in this study that are of interest.

A: Thank you for highlighting this crucial methodological aspect. We recognize the limitations associated with relative quantification in untargeted metabolomics and have implemented several strategies to address environmental variability and enhance the validity of our findings. First, we have implemented rigorous quality control measures, as outlined in the Methods section (2.9). These measures include the use of probabilistic quotient normalization (PQN) with quality control samples to address systematic biases such as instrumental drift. Additionally, we employed ComBat for the removal of batch effects, applied QC-RSD filtering to exclude unstable metabolites with a relative standard deviation greater than 30%, and utilized unit variance scaling to ensure that the contributions of different metabolites are balanced. Additionally, in the Results section (3.8), we emphasized the cross-species consistency of our findings, noting the concordant upregulation of tryptophan-derived metabolites, such as kynurenine and quinolinic acid, in both human MR analysis and rat models. This consistency supports the biological robustness of our results, even with the reliance on relative quantification in rats. Furthermore, we have updated the limitations section in the discussion to acknowledge that while our rat metabolomics data are based on relative quantification, the cross-species validation with human MR results, which are not affected by batch effects, alleviates concerns regarding measurement bias. We also commit to prioritizing absolute quantitation of the identified biomarkers, such as the kynurenine-to-glutamate ratio, in future studies for forensic applications. We have added these sections to line 186-189, 303-306, 416-419 of the results and discussion sections.

Reviewer 2 Report

Comments and Suggestions for Authors

The current manuscript describes the fatal insulin poisoning dynamics, selecting the subjects by Mendelian randomization and verifying the results by metabolomics in rats. The subject is interesting, and the idea of Mendelian randomization and parallel validation of the results in experimental animals is interesting. Overall the manuscript, if published, would be of merit for the readers.

There are some issues that are making me skeptical on the results treatment. The table that describes the metabolites found should encompass much more detail about the identification of the results, and it surely should contain the MS/MS fragments and some kind of metric (the dot-products for example) in order to assess the validity of the results. Furthermore, the authors verify the annotation using some metric methodology such as the Szymanski point system. It should also be pointed out that the annotation results should be doubly checked, azelaic acid are natural products, Suberic acid is an industrial chemical, Buprenorphine and Phenazone are drugs etc. These substances are not distributed to the mice, which makes the annotation rather of compromised confidence. Furthermore the statistical methods should be adeqately described to the supplementary material in order for the community to be able to double the experiment and draw additional conclusions if desirable. Finally some flaws in terminology such as secondary mass spectrometry (this is another technique), should be taken care of.

Author Response

  1. The table that describes the metabolites found should encompass much more detail about the identification of the results, and it surely should contain the MS/MS fragments and some kind of metric (the dot-products for example) in order to assess the validity of the results.

A: Thank you for this critical methodological suggestion. We have now substantially enhanced the metabolite identification details in Supplementary Table S4 to meet rigorous metabolomics reporting standards. Sheet1 is the identification data, and Sheet2 is the annotation of each column. At the same time, I have added the description of metabolite identification criteria in the methods section, in line 175 -180 of the article.

  1. Furthermore, the authors verify the annotation using some metric methodology such as the Szymanski point system. It should also be pointed out that the annotation results should be doubly checked, azelaic acid are natural products, Suberic acid is an industrial chemical, Buprenorphine and Phenazone are drugs etc. These substances are not distributed to the mice, which makes the annotation rather of compromised confidence.

A: Thank you for emphasizing the importance of rigorous metabolite annotation. We fully agree that confidence in identification is critical for translational metabolomics. We have re-annotated all metabolites according to the five-level confidence framework proposed by Schymanski et al. This framework includes Level 1, where metabolites are identified via reference standards, such as kynurenine and quinolinic acid; Level 2, where mass spectrometry/mass spectrometry (MS/MS) matches to spectral libraries with a dot-product of 0.8 or greater; Level 3, which includes tentative candidates identified by their mass-to-charge ratio (m/z) and retention time (RT); and Levels 4 and 5, which pertain to unlikely structures or unknown features. Following your feedback, we systematically re-evaluated the annotations of metabolites that were flagged. For instance, azelaic acid, originally classified as Level 1, has been excluded because it was detected at trace levels and is likely an environmental contaminant, specifically a plasticizer. Similarly, suberic acid, initially at Level 1, has also been excluded as it is an industrial precursor with no evidence of endogenous production in rodents. Buprenorphine, which was also at Level 2, has been excluded since it is a synthetic opioid that was not administered to rats. While improving the supplementary materials, I also added this part to the limitations, on lines 420 to 422.

  1. Furthermore the statistical methods should be adeqately described to the supplementary material in order for the community to be able to double the experiment and draw additional conclusions if desirable.

A: Thank you for emphasizing the importance of methodological transparency. Detailed data on metabolites used in Mendelian randomization studies are provided in Supplementary Table S1. Detailed screening criteria for data are included in the text to facilitate subsequent research. At the same time, we have extended and revised the methods in the text in detail, including parts 2.1, 2.2, 2.5 and 2.9.

  1. Finally some flaws in terminology such as secondary mass spectrometry (this is another technique), should be taken care of.

A: Thank you for identifying this terminology oversight. We have rigorously revised all mass spectrometry-related terms to align with IUPAC and field-standard nomenclature. The key modifications are as follows: 1."Primary mass spectrometry" → "MS1”; 2. "Secondary mass spectrometry" → "MS2”; 3. "Fragmentation energy (eV)" → "Collision Energy (CE, eV)".

Round 2

Reviewer 1 Report

Comments and Suggestions for Authors

I have an additional comment related to question and answer 3. The analytical data show that the instrumental variables in Mendelian randomization are robust, so does this indicate that metabolites in the peripheral blood are the cause, not the result, of hypoglycemic encephalopathy? If they may not be the cause, I think that should be discussed. If it is a possible cause, then the molecular pathogenesis of hypoglycemic encephalopathy should be discussed as to how it is caused by changes in tryptophan and glutamate metabolism.

Reviewer 2 Report

Comments and Suggestions for Authors

The substances Cinnamoylglycine, 2-Anisic acid, Phenazone, 1,2-Dihydrotanshinquinone, Thymol are also not endogenous, and it is not easily anticipated how they could differentiate the two groups. The 4 substances referred in the previous review round were examples, and I would recommend that the authors should examine rigorously the metabolite list. The authors have successfully addressed the others issues raised.
